# Pivotal Role of Phytohormones and Their Responsive Genes in Plant Growth and Their Signaling and Transduction Pathway under Salt Stress in Cotton

**DOI:** 10.3390/ijms23137339

**Published:** 2022-06-30

**Authors:** Irshad Ahmad, Guanglong Zhu, Guisheng Zhou, Xudong Song, Muhi Eldeen Hussein Ibrahim, Ebtehal Gabralla Ibrahim Salih, Shahid Hussain, Muhammad Usama Younas

**Affiliations:** 1Joint International Research Laboratory of Agriculture and Agri-Product Safety of the Ministry of Education of China, Yangzhou University, Yangzhou 225009, China; irshadgadoon737@yahoo.com (I.A.); 006725@yzu.edu.cn (M.E.H.I.); ebtehal201812@gmail.com (E.G.I.S.); 2Key Lab of Crop Genetics & Physiology of Jiangsu Province, Yangzhou University, Yangzhou 225009, China; 3Jiangsu Yanjiang Area Institute of Agricultural Sciences, Nantong 226541, China; xudongsong515@gmail.com; 4Department of Agronomy, College of Agricultural Studies, Sudan University of Science and Technology, Khartoum 13311, Sudan; 5Institutes of Agricultural Science and Technology Development, Yangzhou University, Yangzhou 225009, China; shahid.agro.aaur727@gmail.com; 6Department of Crop Genetics and Breeding, College of Agriculture, Yangzhou University, Yangzhou 225009, China; usamaghias@gmail.com

**Keywords:** phytohormones, abiotic stress, crop improvement, genes

## Abstract

The presence of phyto-hormones in plants at relatively low concentrations plays an indispensable role in regulating crop growth and yield. Salt stress is one of the major abiotic stresses limiting cotton production. It has been reported that exogenous phyto-hormones are involved in various plant defense systems against salt stress. Recently, different studies revealed the pivotal performance of hormones in regulating cotton growth and yield. However, a comprehensive understanding of these exogenous hormones, which regulate cotton growth and yield under salt stress, is lacking. In this review, we focused on new advances in elucidating the roles of exogenous hormones (gibberellin (GA) and salicylic acid (SA)) and their signaling and transduction pathways and the cross-talk between GA and SA in regulating crop growth and development under salt stress. In this review, we not only focused on the role of phyto-hormones but also identified the roles of GA and SA responsive genes to salt stress. Our aim is to provide a comprehensive review of the performance of GA and SA and their responsive genes under salt stress, assisting in the further elucidation of the mechanism that plant hormones use to regulate growth and yield under salt stress.

## 1. Introduction

Cotton (*Gossypium hirsutum* L.) is an economically important crop grown for the textile industry, providing 35% of the total fiber consumption worldwide. China, India, the USA, Brazil, and Pakistan are the major cotton producers in the world [1]. Cotton crops are often exposed to abiotic stresses during growth and development, consequently leading to reduced lint yield and fiber quality [2,3,4]. Salt stress is one of the major challenges limiting world cotton production [5]. The availability of a high content of neutral salts in soils, mainly sodium chloride (NaCl) and sodium sulfate (Na_2_SO_4_), creates salt stress. The negative effects of NaCl on plants are mainly the accumulation of sodium in soils, which reduces water availability and causes toxic effects from the sodium and chloride ions in plants [6]. To cope with salt stress, plants have developed a variety of biochemical and molecular mechanisms, such as selective formation or elimination of salt ions, control over root absorption of ions and transport to leaves parts, regulation of gene expression, stimulation of hormones, etc. [7]. The yield of cotton grown in saline soils can be increased by improving the salt tolerance of cotton [4,8]. The identification of salt-tolerant genes, molecular breeding, and identifying the role of exogenous hormones under salt stress are becoming increasingly important factors for both cotton scientists and growers [9].

Phyto-hormones are small endogenous signaling molecules that are present in plants. Gibberellins, salicylic acid, jasmonic acid, brassinosteriods, ethylene, and humic acid are among the major types of phyto-hormones that are frequently studied in cotton plants [10]. They are directly involved in various physiological and biochemical processes in cotton plants [11,12,13] and, consequently, regulate the growth and development of cotton plants and the yield and quality of cotton fibers [14,15]. In recent years, cotton research has developed rapidly and has led to new discoveries in the hormones involved in cotton growth and development under salt stress. In this manuscript, we have aimed to provide a comprehensive review of the advances in the roles, signaling, and transduction pathways of gibberellins and salicylic acid and the cross-talk communication between them in regulating cotton growth and development under salt stress.

## 2. Gibberellic Acids (GA) Role in Growth and Development under Salt Stress

### 2.1. GA Biosynthesis

Gibberellic acids (also known as gibberellins) are a group of endogenous hormones in plants, consisting of a large group of diterpenoid carboxylic acids. They have been found to play a pivotal role in the whole growth and development of plants, mainly by promoting germination, root growth, stem and leaf elongation, flowering, and boll and fiber development and maturation [13]. To date, a total of 136 various gibberellins have been identified [16], but only a few of them are biologically active [17]. It is known that the accumulation of bioactive GAs is regulated through the modulation of the late steps of GA biosynthesis and catabolism [17,18,19]. In this process, the GA biosynthetic pathway is common to all species during the first steps in which GA 20-oxidase is catalyzed to GA_12_, GA_15_, GA_24_, and GA_9_ (Figure 1). Furthermore, GA_9_ is then converted by GA_3_-oxidase to GA_4_ [20]. Similarly, GA9 is further converted to GA_4_ by GA-3 oxidase [20].

Numerous reports have demonstrated that GAs are involved in certain biological processes and promote the vegetative and reproductive growth of crops in response to stresses [21,22,23,24]. The exogenous application of GA in cotton can enhance plant growth and fiber development [15,25]. Additionally, it can increase the content of indoleacetic acid (IAA) and abscisic acid (ABA) during fiber growth and, consequently, improve strength, micronaire reading, and maturation in the natural color of cotton fiber (Figure 1). In several studies, salt stress up-regulated those genes involved in the GAs activation, such as the *GA2ox7* gene, and the suppression of GA signaling (DeLLA proteins encoding genes) in *Arabidopsis* plants [25,26,27,28,29,30]. However, little information is available about the effects of salt stress on the biosynthesis of active GA_4_ [31]. In this sense, further investigation is required to better understand the effects of GA on cotton fiber growth and its relation with IAA and ABA content in cotton plants under salt stress conditions [32].

### 2.2. Gas Signaling Transduction

The activation of the signaling pathways of GA and its functions occur due to the interaction between the DeLLA proteins and GA-intensive dwarf1 (GID1). DeLLA proteins are the main negative regulators of GAs, playing a vital role in regulating other plant hormones and signal pathways under salt stress [33]. GID1 is a receptor binding to GAs [26,32,34]. The combination of GID1 with active GAs helps in the transmission of GA signals to the DeLLA proteins to regulate the expression of the synthesis genes of GA [13]. The degradation of DeLLA proteins has been identified to improve plant survival under stress conditions via reducing the accumulation of reactive oxygen species (ROS) [35]. The over-expression of GID1 possibly increases the sensitivity of plants to GAs [36]. Meanwhile, the DeLLA proteins target the *XERICO* gene, which is involved in regulating the plant responses to salt stress [36]. DeLLA proteins also enhance the plants responses to salt stress by inhibiting the synthesis of GA through XERICO induction [37]. In a previous study, the content of GA in *Arabidopsis* thaliana was reduced, while the accumulation of negative regulatory factor DeLLA proteins was increased under salt stress [38] (Figure 1). To date, the detailed mechanism of DELLA proteins targeting the XERICO gene is far from clear.

### 2.3. Seed Priming with GA Mitigates Salt Stress

Soaking seeds with a suitable concentration of GA can protect seed deterioration and overcome the negative effects of salt stress (osmotic stress, ion toxicity, and nutrients imbalance) (Figure 2). The reports of [39] for maize and in [40] for sorghum demonstrated that soaking seeds with GAs facilitated their germination and increased seedling length, as GAs stimulated cell division and cell elongation. Seeds treated with GA_3_ at 50 or 100 ppm had a higher accumulation of potassium in the vegetative parts of wheat plants [41]. An addition of 0.5 mg/L of GA_3_ played an important role in increasing the weight of the dry seedling. Compared to unsoaked seeds, soaked seeds with 100 mg/L of GA_3_ significantly increased in chlorophyll content and had a lower ratio of sodium to potassium in maize. Seeds soaked with 300 mg L^−1^ of GA_3_ in maize had the highest chlorophyll content in the seedling leaves and the highest dry seedling weight during the fall and spring seasons as compared to non-soaked seeds [42]. Similarly, rice seeds treated with GAs had a significantly higher vitality and vigor in the seedlings [43]. Hamdia and Shahdad [44] observed that the maize plants treated with GA_3_ had a higher chlorophyll content and a lower movement of Na^+^ in the plants. Numerous research studies have reported that seeds treated with GAs had an improved plant tolerance to salt stress, but the knowledge surrounding the application of GA to cotton seeds is inadequate and needs to be further investigated.

## 3. Role of **Salicylic Acid (SA) in Plant Growth and Development** of Cotton **under Salt Stress**

### 3.1. SA Biosynthesis

The biosynthesis of SA can be found in different biochemical pathways, such as isochorismate synthase (ICS) and phenylalanine ammonia-lyase (PAL). It is synthesized from chorismite, and this process takes place in the chloroplast and cytosol [45,46]. These enzymes are considered the key regulators of SA functions and are known to be balanced by different environmental stresses [47]. The ICS pathway is involved in the biosynthesis of SA and the responses to different crop diseases caused by pathogens. An understanding of the role of the ICS pathway in plant growth from germination to maturity is lacking and needs to be further investigated, especially under salt stress [48,49,50]. Moreover, it has been identified that the PAL pathway is essential in rice, while the ICS pathway seems to be more essential for SA accumulation in *Arabidopsis* [46]. However, higher accumulations of SA in cotton plants through these pathways still needs to be investigated. Various research has shown that SA biosynthesis regulation can be different within different parts of the plant. For example, in rice, the SA concentration in the shoots is much greater than in the roots [51,52]. SA can regulate the closing of stomata through SA-induced protein kinases, including cyclic adenosine monophosphate (cAMP) and cyclic guanosine monophosphate (cGMP) [53,54]. cAMP and cGMP are the secondary messengers and regulate many physiological activities in plants, including gene expression and cell cycle maintenance. However, the metabolic functions regulated by SA is far from fully understood (Figure 3).

### 3.2. SA Signaling Transduction

SA is an essential signaling molecule in plants and plays a vital role in regulating various responses to abiotic stresses, including salinity, heat stress, ozone exposure, and heavy metals [55,56]. For SA biosynthesis, two signaling transduction pathways have been identified in plants, including PAL and ICS [57]. In tobaccos, the silencing of PAL genes, and in *Arabidopsis*, the chemical constraint of PAL activity decreases SA accumulation [58]. In the ICS pathway, plants produce SA from chorismate via isochorismate due to ICS activity in *Arabidopsis* [59]. The occurrence of SA accumulation in cotton via these pathways under salinity stress need to be further investigated for regulating crop improvement. SA in plants is produced by glucosylation, methylation, or hydroxylation in various forms and transferred to different parts of the plants to improve crop yield [60]. The combination of SA with jasmonic acid (JA), ethylene (ET), auxin, gibberellic acid (GA), and abscisic acid (ABA) is considered to constitute a portion of the signaling network using exchange talk and linkage between these phyto-hormones [61].

### 3.3. Application of SA Mitigates Salt Stress

SA, a type of phenolic phyto-hormone, plays a critical role in many biochemical and physiological processes in plants, such as photosynthesis, nitrogen metabolism, proline metabolism, and antioxidant activity [47,62,63,64,65,66,67]. Various studies on different crops have shown that SA can improve plant resistance to major abiotic stresses, including salinity [47,68]. SA was applied to stressed plants via different methods such as seed soaking, irrigation, spraying, or being adding to the nutrient solution, suggesting that SA induced an abiotic stress tolerance mechanism [69,70,71,72,73,74]. It has been demonstrated that a reasonable exogenous application of SA can enhance the growth and photosynthesis activity of cotton seedlings under salt stress and help the plants to withstand stressful conditions [75]. Similarly, an exogenous application of SA enhanced the growth performance and photosynthetic activity and simultaneously reduced oxidative stress in mung bean (*Vigna angularis* L.) plants under salt stress [76]. Mutant sid2 seed germination was highly susceptible to salt stress, but a supply of physiological SA decreased the negative effects on seed germination under salt stress [77]. Maize and pea plants treated with SA showed an alteration in antioxidant activities [78,79], but no stress symptom was exhibited in wheat plants for certain antioxidant activities [80]. Salt stress reduced SA concentration in *Dixie iris* plants [81]. Similarly, SA concentration was reduced in tomato [82] and soybean [83] plants under salt stress. The suitable range of SA in most plants has been identified as ranging from 0.1 mM to 0.5 mM [56]. Higher or lower concentrations of SA make the plant more susceptible to various stresses [54]. In *Matricaria chamomilla*, 50 μM SA promoted plant growth while 250 μM SA inhibited plant growth [84]. Additionally, a 0.5 mM concentration of SA promoted photosynthesis activity and growth in *Vigna radiata* but 0.1 mM SA hindered it’s growth [85]. Seeds primed with SA can be used as necessary tools for improving major GSH-based (glutathione) H_2_O_2_-metabolizing enzymes such as the GST (glutathione S-transferase) gene family [47].

Plant varieties, treatment period, age, and plant organ treatment can influence the role of SA in plants [63,86]. One of the molecular studies demonstrated that SA is maintained in plants at the gene level and thus can enhance abiotic stress tolerance in plants [87]. It has been reported that SA induces many of the genes responsible for heat shock proteins (HSPs), encoding chaperones, antioxidant enzymes, and secondary metabolites, including sinapyl alcohol dehydrogenase (SAD) and cinnamyl alcohol dehydrogenase (CAD) [88]. In tomato plants (*Solanum lycopersicum*), SA application helped the plant cope with salt stress injury by causing characteristic changes in the expression pattern of the GST gene family, including *SlGSTT2*, *SlGSTT3*, and *SlGSTF4* (Figure 3). Similarly, an exogenous supply of SA interestingly enhances salt tolerance in wheat (*Triticum avestivum* L.) because of the improved transcriptional level of antioxidant genes, including *GPX1*, *GPX2*, *DHAR*, *GR*, *GST1*, *GST2*, *MDHAR*, and *GS* [89]. Due to the accumulation of hydrogen peroxide (H_2_O_2_), SA may expose the plants to oxidative stress [89,90,91]. But in the case of salt stress, the effects are rather vague [54,76]. SA can accelerate the production of ROS (reactive oxygen species) in photosynthetic tissues in plants during salt stress, therefore playing an important role in the growth of stress symptoms. In *Arabidopsis* plants, the SA accumulation mutant snc1 boosted salt-induced injury; at the same time, the SA-deficit mutant nahG boosted SA signaling, blocking mutants *npr1-1* and *snc1* and, as a result, reduced salt injury [92] (Figure 3). The performance of SA in strengthening salinity resistance mechanisms in plants has been widely identified in many crop species such as *Brassica juncea* L. [68,85] and alfalfa (*Medicago sativa* L.) [73], yet little information surrounding the detailed mechanism in cotton is up-to-date.

## 4. GAs and SA Cross-Talk during Salt Stress

Currently, salt stress is one of the major abiotic stresses affecting crop growth and production [54,87,93,94]. Salt stress mainly creates oxidative stress, ion toxicity, and nutritional imbalances in various crops (Figure 4). To mitigate the negative effects of salt stress, antioxidant enzymes play a role in neutralizing uncontrolled ROS over-production in various subcellular compartments [95]. At the beginning of antioxidant defense, signaling molecules GA and SA cope with salt stress and modulate many morpho-physiological activities [96]. There are previous researches indicating that GAs have an active performance in SA biosynthesis and action [97]. The exogenous supply of GA_3_ or over-expression of *GASA* genes reverts to salt stress, which is regulated by SA biosynthesis in *Arabidopsis* because *GASA* genes are the putative intermediates between GA and SA hormones (Figure 4). *GASA* genes, a family of GA-induced genes, play a vital role in plant response to various stresses [98,99]. During seed germination, FsGASA4 transgenic lines are more resistive to salt stress in *Arabidopsis* [100]. When compared with the wild type of seed, the FsGASA4 transgenic lines seeds had a higher SA concentration. In the presence of a GA_3_ supplemented medium, the FsGASA4 seedlings enhanced the expression of isochorismate synthase (*ics1*) and the natriuretic peptide receptor (*npr1)*, which is involved in SA biosynthesis and action [100]. The GASA genes play an indispensable role in plant growth and coping with salt stress by regulating SA biosynthesis. Therefore, *GASA* genes and their association with SA biosynthesis could constitute a potential strategy for improving cotton production under salt stress in near future.

## 5. GAs and SA Regulating Morph-Physiological Activity

GAs and SAs play a variety of physiological roles in plant growth and development. GA especially enhances germination, leaf expansion, starch metabolism, and cell enlargement and is involved in regulating flowering [101,102]. GAs also trigger stem cell elongation, cell division, and hyper elongation and, as a result, increase plant height [103]. Various enzymes such as amylase are encouraged by GA for germinating cereal grains and inducing maleness in dioecious flowers [103]. SA enhances plant growth, fruit ripening by regulating chlorophyll pigments (*a*, *b*, and carotenoids), chloroplast structure maintenance, and regulates stomata closure and Rubisco activity [104,105]. Exogenous SA promotes protein synthesis, which is necessary for the degradation and mobilization of seed proteins during germination [106]. A variety of studies indicate that GAs and SA are the master regulators of plant growth and physiology during abiotic stresses [104]. Growing evidence has shown that miRNAs, GAs, and SA are coordinated in regulating plant physiology. Several components in SA and GA pathways are targets of specific miRNAs, which play an indispensable role in plant growth and physiology, such as meristem function, organ morphogenesis and different stress response [107,108]. Previous studies demonstrated that Sly-mirR159-SIGAMYB2 and Sly-miR171-SlGRAS24 pathways control the morph-physiological activity of tomato fruit by regulating GA biosynthesis [109]. The Sly-miR159 promoted GA biosynthesis occurs via the direct repression of GA biosynthetic gene *SlGA3ox2* via SlGAMYB2. Sly-mirR159 and Sly-miR171, through their targeted gene, *SlGRAS24*, regulate root length, plant height, fruit growth, and development [109,110]. The down-regulation of Sly-mirR159 and over-expression of SlGAMYB2 resulted in larger fruits with increased length and width [109]. Similarly, the down-regulation of Sly-mirR160 enhanced fruit length while decreasing fruit width [111], and the over-expression of Sly-miR156 leads to abnormal carpel and fruit shape [112]. The performance of these miRNAs in cotton morph-physiological activity under salt stress is not known and needs to be further investigated.

## 6. Role of GA and SA Synthase Genes in Salt Stress Tolerance

The interference of GA and SA biosynthesis genes with signaling transduction genes in mitigating salt stress is shown in Table 1. The GA2 oxidase (*GA2ox*) gene affects plant metabolism and the photosynthesis process, which are important for plant survival in adverse conditions. The *GA2ox* genes make plants more resistant to salt stress. *GA2ox* is a class of oxidase and plays a vital role in regulating the synthesis of GA [113]. Previous research demonstrated that *GA2ox* helps in the conversion of bioactive GAs to an inactive form [114]. Therefore, *GA2ox* is suggested as one of the main characters in affecting the content of bioactive GAs [115]. *GA2ox* genes have been characterized in various plants, such as rice [116], potato [117], poplar [118], breadfruit [114], and jatropha [119]. It has been reported that a high level of GA2ox expression is associated with a low concentration of bioactive GAs [115,120]. The reduction in GA in dwarf and delayed flowering (DDF1) in *Arabidopsis* is due to the increased expression of the GA 2-oxidase 7 gene (*GA2ox7*), which encodes a C20-GA deactivation enzyme [29]. In rice, the performance of GA during salt stress is due to the over-expression of *GA2ox5* genes. Previous research showed that the overly-expressed GA2ox5 genes are capable of coping with high salinity [121]. The adaptation to salt stress is encouraged by decreasing GA levels caused by the induction of GA 2-oxidase genes [97]. In *Arabidopsis thaliana*, the over-expression of *AtGA2ox7*/*AtGA2ox8* genes reduced the content of GAs along with delaying the flowering time. In transgenic rice plants, the expression of *GA2ox6* genes increased the grain yield by 10–30% during biotic stresses [116]. Various studies constitute well-documented information on these genes in other crops, but the role of these gene members under salt stress in cotton is still vague and could be further investigated for overall crop improvement.

A salt-tolerance mechanism named Tudor staphylococcal nuclease (TSN) is involved in the regulation of gene expression. TSN is involved in stress conditions and role ribonucleoproteins (RNP) and transcriptional activity [128,129]. TSN modulates GA20ox3 mRNA, which is considered to be key for GA-biosynthesis enzymes. A study with two RNAi lines (TSN1, TSN2) and mutant GA20ox3 indicated that the two RNAi lines, TSN1 and TSN2, and the GA20ox3 mutant showed growth damage during salt stress. The levels of GA20ox3 mRNA increased in TSN1 and decreased in TSN2, showing that TSN regulated the mRNA levels of GA20ox3. The exogenous application of GA_3_ partially recovered the growth damage of the two RNAi lines and the GA20ox3 mutants, evidencing that decreased GA levels and increased DELLA accumulation are responsible for growth hindrance. In *Arabidopsis*, higher levels of GA20ox3 have accumulated at 150 mM NaCl level. The induction of GA20ox3 is necessary for plant growth under salt stress [130]. TSN is the best process for recognizing the pathways of gene expression and coping with salt stress. However, knowledge about the TSN process in cotton for various genes expression is lacking and needs to be further investigated.

Furthermore, the over-expression of *GASA14*, a gene member of the GA-stimulated *GASA* genes family, enhanced salt resistance in *Arabidopsis* transgenic plants [124]. When compared to wild-type plants, the *GASA14* mutant’s deficit plants showed more growth imperfections. The *GASA14* mutants accumulated higher concentrations of (Hydrogen peroxide) H_2_O_2_, while lower concentrations of H_2_O_2_ had accumulated via the over-expression of the GASA14 line. *GASA14* genes showed stress resistance in crops by modulating ROS accumulation [124].

Moreover, the *Lhc* genes family, particularly *Lhcb2* and *Lhcb1*, in *Arabidopsis* has been widely studied [131], but most of the *Lhc* family genes are still unknown in cotton thus far [125]. The previous research showed that cotton tetraploid species (*Gossypium hirsutum* and *Gossypium barbadense*), and their two ancestral diploid species (*Gossypium raimondii* and *Gossypium arboreum*), have been frequently used to identify the cotton genes family [24,132,133,134]. Among these four cultivated species, *Lhc* family genes, represented as *CAB* genes, were some of the first plant genes to be sequenced [25,135].

Investigations have shown that the *Lhc* family genes play a pivotal role in the functioning of GhLhc family proteins. In various crops, research on *Lhc* family genes, such as *Arabidopsis thaliana*, Algae, and tea, showed that these proteins play a pivotal role in plant light protection [122,126,136]. For plant growth, development, and photosynthesis, light is very important, and the Lhc family proteins are necessary components and key factors affecting cotton growth and yield [15,137]. The light-harvesting chlorophyll a/b binding (Lhc) super-family proteins play an essential role in capturing light from the sun as well as in photo-protection under stress conditions [138]. The Lhc family consists of two evolutionary groups, namely Lhca and Lhcb, that are closely linked with photo-systems PSI and PSII [139]. The study on PSI and PSII showed that Lhc family proteins closely associated with PSI and PSII are very important in light capturing [140]. A study showed that Lhc family proteins are involved in the regulation and supply of excitation energy between PSI and PSII in photo-protection, as well as in response to salt stress [141,142,143]. Photosynthesis in taller plants utilizes chlorophyll and carotenoid content during salt stress to improve growth and cope with salt stress [144]. Therefore, studies on many of the *Lhc* family genes and their relationship with chlorophyll pigments in *Gossypium hirsutum* L. are still lacking [125].

*GhPHDs* genes are involved in regulating cotton growth and development, particularly ovule and fiber development. It has been noted that *GhPHDs* genes (especially *GhPHD5*, *GhPHD80*, and *GhPHD88*) respond well to various phyto-hormone signal transduction pathways and enhance cotton tolerance to different stresses, including salt, heat, and drought [145]. Wu et al. [145] showed that phyto-hormones improved plant tolerance to abiotic stresses via *GhPHD* genes and their co-factors, but their regulatory mechanisms and interaction network still need to be further investigated. The expression *GhPHD5* genes has been significantly increased after treatments of MeJA (Methyl jasmonate), IAA (Indole-3-acetic acid), and BL (brassinolide). Previous studies demonstrated that *GhPHD5* showed higher expression under SA application. During SA treatment, *GhPHD40* genes are significantly up-regulated, showing that *GhPHD840* positively respond to SA signals. Likewise, after GA treatment, *GhPHD80* and *GhPHD88* genes also undergo a significant increase. A study showed that *GhPHD5* positively interacts with slow motion (SLOMO) proteins and is involved in the regulation of the auxin signal transduction pathway, facilitating seed germination and organ formation to regulate plant growth and development [146]. Correspondingly, *GhPHD18* is also positively co-expressed with highly hydrophilic proteins that regulate flowering locus C (FLC), which affects the flowering time of meristem [147]. *GhPHD* genes improve cotton growth and yield by regulating the signal transduction pathway of auxin, but the performances of these genes in regulating other phyto-hormone signal transduction pathways requires further investigation.

## 7. Conclusions and Future Directions

Salt stress is considered to be a major threat to crop production. To overcome the adverse effects of salt stress, plants use various physiological and biochemical processes and molecular mechanisms [125]. The current investigation anticipates that phyto-hormones, such as GA and SA, have strong potential as tools for reducing or mitigating the negative effects of salt stress in cotton plants. In addition, seed priming with GA and SA improves seed germination, plant physiology, and antioxidant capacity and, as a result, plant growth and development during salt stress. The exogenous application of these hormones has been shown to be essential for plant growth and development under salt stress. Phyto-hormones are involved in fiber growth and development [14], but the performance of these hormones under salt stress needs to be further investigated [10].

Moreover, cross-talk studies between GA and SA and their responses to abiotic stresses make for attractive goals in molecular research. The synergy between GA and SA regulates many morpho-physiological activities during salt stress due to the *GASA* genes family. Despite plenty of research literature available, the following issues remain for further investigation.

The positive effect of the exogenous application of GAs has been confirmed, evidenced by increasing IAA and ABA during cotton fiber growth and strengthening, and increasing the micronaire and maturation of color. However, it is important to elucidate the role of GA hormones and their relations with the IAA and ABA contents under salt stress.

It has been confirmed that the accumulation of bioactive GAs is regulated via the modulation of the last steps of GA biosynthesis and catabolism. The process of the GA biosynthetic pathway is common to all species during the first step, in which GA 20-oxidase catalyzes the GA_12_, GA_15_, GA_24_, and GA_9_ processes, with GA_9_ then converted into GA_4_ by GA 3-oxidase. However, there exists little up-to-date information surrounding the detailed mechanism of bioactive GA in cotton and in various crops that facilitate plant growth and yield under salt stress.

DELLA proteins target the *XERICO* gene, which plays a vital role in plant growth. It has been confirmed that DELLA enhances the plant response to salt stress by inhibiting GA synthesis via XERICO inductions. However, the detailed DELLA protein targeting of the *XERICO* gene mechanism is far from being clearly understood.

Under environmental stress conditions, crop yield is reduced due to the fluctuations in GA and DELLA levels in plants. Further studies are needed to understand the physiological and molecular mechanisms of GA and DELLA protein fluctuation when foreign GA is applied.

SA biosynthesis pathways vary in different crops. For example, the PAL pathway is essential in rice, while the ICS pathway seems to be more essential in *Arabidopsis*. However, a suitable pathway for cotton under salt stress is unknown.

SA regulates the closing of stomata via SA-induced protein kinases cAMP and cGMP. These are the secondary messengers and regulate many physiological activities such as gene expression, cell cycle maintenance, and the metabolic functions of the plants. However, more studies are needed to investigate these secondary messengers during salt stress.

The positive role of SA has been widely studied in different cereals and cash crops. However, contrasting the functions and mechanisms of some SA biosynthesis genes with those of the *GASA* genes (the family of GA-induced genes) needs to be further studied.

GA and SA synthesis genes play an important role in regulating cotton growth and yield. Additionally, many genes affect plant metabolism and photosynthesis processes, which are important for plant survival during salt stress. For example, *GA2ox* genes improve grain yield by 10–30% in rice under salt stress. An understanding of the performance of these GA synthesis genes within cotton is lacking. Similarly, *Lhc* family genes play a functioning role in Ghlhc family proteins, which in turn play a vital role in the plant’s protection against light. For plant growth and development, light is a very important factor during photosynthesis. Therefore, it is important to elucidate the roles of different *Lhc* genes in cotton under salt stress.

It has been confirmed that *GhPHD* genes improve cotton growth and yield by regulating the signal transduction pathway of auxin. However, there is a lack of understanding of the performance of these genes in the regulation of other phyto-hormone signal transduction pathways.

## Figures and Tables

**Figure 1 ijms-23-07339-f001:**
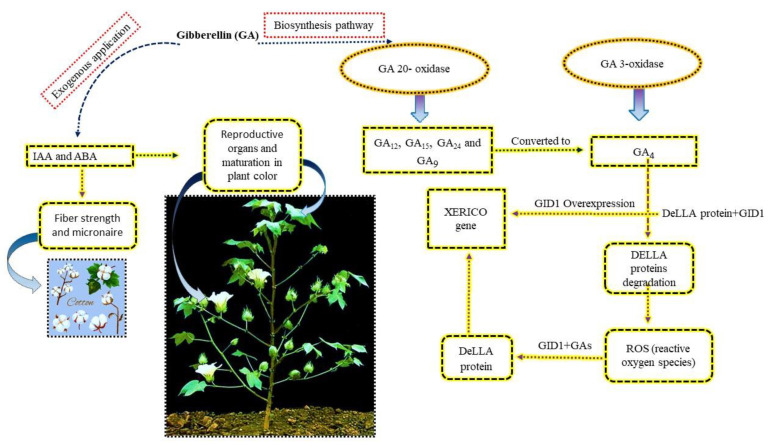
The regulatory role of an exogenous supply of GA and its biosynthetic pathway in mitigating salt stress. The GA 20-oxidase catalyzes GA_12_, GA_15_, GA_24_, and GA_9._ Similarly, GA_9_ is further converted to GA_4_ by GA-3 oxidase. The GA signaling activation pathway occurs due to the interaction between DeLLA proteins and GID1, which leads to the degradation of DELLA proteins. The degradation of DELLA proteins reduces the accumulation of reactive oxygen species (ROS) and improves plant growth under salt stress. The over-expression of GID1 increases plant sensitivity. GID1 with GA_S_ carries the GA signal to the DELLA proteins to modulate the expression of GAs synthesis genes, XERICO. XERICO genes reduce salt stress and improve crop growth and yield. GA exogenous application enhances the contents of indole-3-acetic acid (IAA) and abscisic acid (ABA) and, as a result, improves fiber strength, micornaire reading, reproductive organ development, and maturation in plant color.

**Figure 2 ijms-23-07339-f002:**
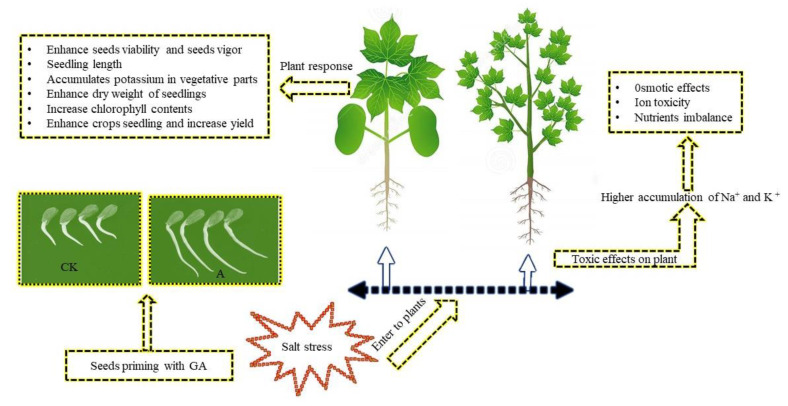
The regulatory roles of seed priming with GA in mitigating salt stress. Salt enters the plants through the roots, which creates toxic effects on the plants due to the accumulation of sodium (Na^+^). The higher availability of Na^+^ in soils makes the plants more susceptible to osmotic pressure, ion toxicity, and nutrient imbalances. Seed priming with GA before sowing can protect the seeds from germination through to maturity and enhance the crop yield.

**Figure 3 ijms-23-07339-f003:**
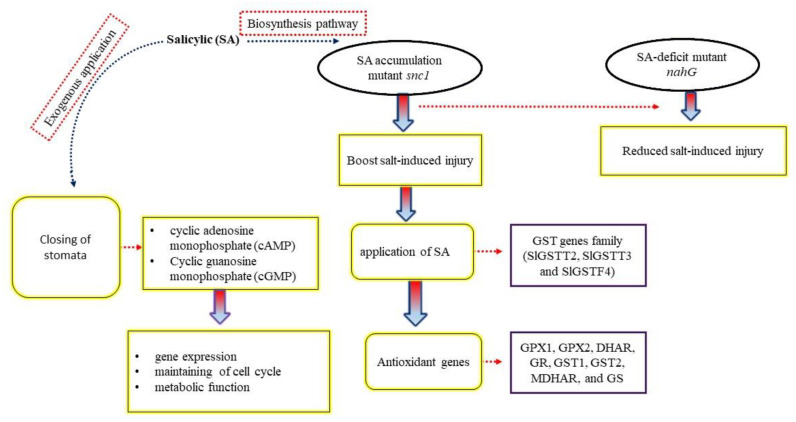
Illustration of the vital performance of an exogenous supply of SA and its biosynthetic pathway in mitigating salt stress. SA accumulation mutant *snc1* triggers salt-induced injury, meanwhile SA deficit mutant *nahG* boosts SA signaling, blocking *npr1*-*1* and *snc1* and as a result, reduces salt stress. Due to the application of SA, salt-induced injury decreases by the variation in the expression pattern of the GST family of genes, such as *SIGSTT2*, *SIGSTT3*, and *SIGSTT4* and antioxidant genes including *GPX1*, *GPX2*, *DHAR*, *GR*, *GST1*, *HST2*, *MDHAR*, and *GS*. During salt stress, SA application regulates stomata closure via secondary messengers (adenosine monophosphate (cAMP) and cyclic guanosine monophosphate (cGMP)) and these secondary messengers further regulate a variety of physiological activities and gene expressions, maintaining cell cycles and metabolic functions of the crops.

**Figure 4 ijms-23-07339-f004:**
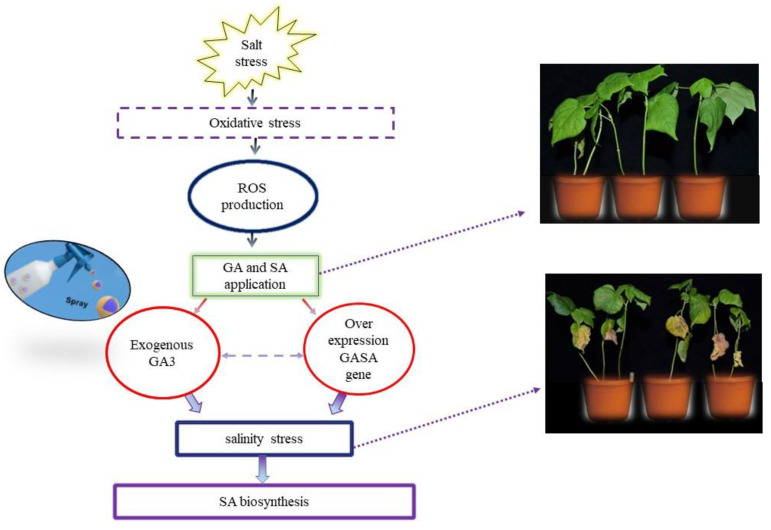
The role of gibberellin and salicylic acid cross-talk in mitigating salt stress in cotton. Salt stress induces oxidative stress. Oxidative stress triggers the production of reactive oxygen species (ROS) and causes cell death. ROS production is reduced by the supply of exogenously applied GA and SA. The overexpression of *GASA* genes and the exogenous supply of GA_3_ at the same time reverse salt stress which is further regulated by SA biosynthesis in plants due to the presence of *GASA* genes.

**Table 1 ijms-23-07339-t001:** Interference of gibberellic acids (GAs) and salicylic acid (SA) biosynthesis and signaling transduction genes in mitigating salt stress in various crops.

GA and SA	Crops	Genes	Genes Response to Salt Stress	References
Biosynthesis genes	Cotton	*GA2ox*	increased salinity resistance	[13]
	Rice	*GA2ox*, *GA2ox5*, *GA2ox6*	increased salinity resistance	[116,121]
	Potato	*GA2ox*	increased salinity resistance	[117]
	Poplar	*GA2ox*	increased salinity resistance	[118]
	Breadfruit	*GA2ox*	increased salinity resistance	[114]
	*Arabidopsis thaliana*	*AtGA2ox7/AtGA2ox8*, *GA2ox7*, *GASA*, *Lhc*	increased salinity resistance	[122,123]
	*Arabidopsis thaliana*	*gasa14* mutant’s	Reduced salinity resistance	[124]
	Cotton	*Lhc (CAB)*	increased salinity resistance	[125]
	Tea plants	*Lhc*	increased salinity resistance	[126]
Signaling transduction genes	*Arabidopsis thaliana*	*XERICO*	increased salinity resistance	[37]
	*Arabidopsis*	*GASA* overexpression or exogenous supply	reduced salinity resistance	[97]
	Tomato	*GST* (*SlGSTT2*, *SlGSTT3*, and *SlGSTF4*)	increased salinity resistance	[127]
	Wheat	*GPX1*, *GPX2*, *DHAR*, *GR*, *GST1*, *GST2*, *MDHAR*, and *GS*	increased salinity resistance	[89]

## Data Availability

Not applicable.

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
