# Peer review of "Pivotal Role of Phytohormones and Their Responsive Genes in Plant Growth and Their Signaling and Transduction Pathway under Salt Stress in Cotton"

_ijms, 2022, doi:10.3390/ijms23137339_

Round 1

Reviewer 1 Report

The  authors herein provider a picture of the activity of two hormones in plants. They descrive pathway and signalling including also downstream genes. However there are some issues that news that need ti be addressed. Growing evidence has shown that miRNAs and SA and GAs are coordinated regulating plant physio. Several components in SA and GA pathways are targets of specific miRNAs, and as well SA and GA also modulate post-transcriptional regulation via the expression of selected miRNA genes. 

Thus, i strongly suggest to add at least a new paragraph focusing on this aspect.

Author Response

Responses to the Reviewers and Editors

Dear editors and reviewers,

The revised manuscript entitled “Pivotal role of phytohormones and their responsive genes in plant growth and their signaling and transduction pathway under salt stress in cotton” (Manuscript ID IJMS-1785654), has now been uploaded on the manuscript tracking system. We have considered all the comments and suggestions and corrected the manuscript or added clarifying text to meet almost all of the recommendations of the reviewers. The following is the item-by-item lists of our responses to the comments made by each reviewer:

Responses to Reviewer 1

Q1: The authors herein provide a picture of the activity of two hormones in plants. They describe pathway and signaling including also downstream genes. However, there are some issues that news that need to addressed. Growing evidence has shown that miRNAs and SA and GAs are coordinated regulating plant physio. Several components in SA and GA pathways are targets of specific miRNAs, and as well SA and GA also modulate post-transcriptional regulation via the expression of selected miRNA genes. 

Thus, I strongly suggest to add at least a new paragraph focusing on this aspect.

A1: Thank you very much for your valuable suggestion. I already added one paragraph in the revised manuscript according your suggestion. This paragraph is important and it makes the manuscript more informative. The paragraph is enlisted in the review paper is on number “5” with red color highlight. If you want us to do more modifications in the manuscript, please feel free to let me know. Thank you once again  

Reviewer 2 Report

This is a comprehensive review about the role of phyto-hormones gibberellic acid (GA) and salicylic acid (SA) in seed germination, photosynthesis and plant growth under salt stress. Authors examine exogenous application of these hormones, their responsive genes and usually beneficial effect for yield in cotton. The present knowledge about the biosynthesis pathways for GA and SA are also reviewed. 

General comments: 

Pages and lines should be numbered to expedite the review process.

The English is mostly good, but there are some sentences that need to be improved.

The reference list is comprehensive. 

Two lines into section 3.1: “phenylalanine ammonia-lyase (PLA)” should be “(PAL)”

There seems to be some repetition between sections 3.1 and 3.2, which could be streamlined. 

Figures: these are informative. The changing colour of the box backgrounds makes the letters difficult to see. A uniform background would be better. 

A part of Fig. 2 seems to be cut off. 

Fig. 4 caption:  “Salt stress induces oxidative stress which triggers the production of reactive oxygen species (ROS) and causes salinity.”??? ROS does not cause salinity.

Author Response

Responses to Reviewer 2

Q1: Pages and lines should be numbered to expedite the review process.

A1: All the pages and lines have been already numbered.

Q2: The English is mostly good, but there are some sentences that need to be improved.

A2: Thank you very much. The manuscript has now been polished by an English-native professional. Grammatical errors have now been corrected to the largest extent.

Q3: The reference list is comprehensive. 

A3: We have now optimized the list of the references. We will certainly cooperate with the reviewers and editors upon further requests.

Q4: Two lines into section 3.1: “phenylalanine ammonia-lyase (PLA)” should be “(PAL)”

A4: Revised as suggested. Many thanks for your careful revision.

Q5: There seems to be some repetitions between sections 3.1 and 3.2, which could be streamlined. 

A5: Thank you very much, In the 3.2 section the similarity between these two paragraph has been removed with some modification. 

Q6: Figures: these are informative. The changing color of the box backgrounds makes the letters difficult to see. A uniform background would be better. 

A6: Good suggestion. Revised as suggested.

Q7: A part of Fig. 2 seems to be cut off. 

A7: Thanks. Figure 2 has been fixed now.

Q8: Fig. 4 caption: “Salt stress induces oxidative stress which triggers the production of reactive oxygen species (ROS) and causes salinity.”??? ROS does not cause salinity.

A8: Fig. 4. has been changed with some minor modification. Yes, you are right. We missed the oxidative stress portion in figure that’s why it creates misunderstanding. Salt stress induced oxidative stress and as a result ROS production arises and the higher production of ROS causes cell death. Thank you for correction.

Thank you very much for the time and consideration. We hope the changes made to the manuscript and the above explanations are adequate and clear, and that this manuscript can now be acceptable for publication in the journal of International Journal of Molecular Sciences. We will certainly remain fully cooperative with your further comments and suggestions and make all necessary changes in the future.

Yours sincerely,

Dr. and Prof. Guisheng Zhou

Joint International Research Laboratory of Agriculture and Agri-Product Safety of the Ministry of Educa-tion of China, Yangzhou University, Yangzhou 225009, Jiangsu Province, China.
